# Retroviral Infection and Commensal Bacteria Dependently Alter the Metabolomic Profile in a Sterile Organ

**DOI:** 10.3390/v15020386

**Published:** 2023-01-29

**Authors:** Jessica Spring, Vera Beilinson, Brian C. DeFelice, Juan M. Sanchez, Michael Fischbach, Alexander Chervonsky, Tatyana Golovkina

**Affiliations:** 1Committee on Microbiology, University of Chicago, Chicago, IL 60637, USA; 2Department of Microbiology, University of Chicago, Chicago, IL 60637, USA; 3Chan Zuckerberg Biohub, San Francisco, CA 94158, USA; 4Department of Bioengineering, Stanford University, Stanford, CA 94305, USA; 5Department of Pathology, University of Chicago, Chicago, IL 60637, USA; 6Committee on Immunology, University of Chicago, Chicago, IL 60637, USA; 7Committee on Genetics, Genomics and System Biology, University of Chicago, Chicago, IL 60637, USA

**Keywords:** retroviruses, microbiota, commensal bacteria, metabolites

## Abstract

Both viruses and bacteria produce “pathogen associated molecular patterns” that may affect microbial pathogenesis and anti-microbial responses. Additionally, bacteria produce metabolites, while viruses could change the metabolic profiles of the infected cells. Here, we used an unbiased metabolomics approach to profile metabolites in spleens and blood of murine leukemia virus-infected mice monocolonized with *Lactobacillus murinus* to show that viral infection significantly changes the metabolite profile of monocolonized mice. We hypothesize that these changes could contribute to viral pathogenesis or to the host response against the virus and thus open a new avenue for future investigations.

## 1. Introduction

The intestinal commensal microbiota is a key factor that mediates host health by providing nutrients and vitamins, supporting the development and shaping of the secondary lymphoid organs in the intestine, and conferring resistance of colonization by pathogenic microorganisms [1]. Many of these and other microbiota-dependent processes are mediated by a number of commensal bacteria-derived high- and low-abundance molecules. Various commensal small molecules exhibit potent biological activities that can influence the host’s cellular activities [2,3,4]. Of these small molecules, metabolites have been shown to serve as an energy source, modulate diseases and psychiatric disorders, promote intestinal barrier function, and modulate the immune system [5,6,7].

Metabolites have both positive and negative effects on pathogenic bacteria. These effects include enhanced biofilm formation, increased expression of virulence factors [8,9,10], disruption of bacterial cell structures, suppression of bacterial growth, and stimulation of innate immune cell proliferation [8,11,12,13,14]. Viral infections are also impacted by metabolites. Metabolites can be used as energy sources to promote viral replication [15,16], whereas other metabolites promote the immune response and augment interferon (IFN) expression in a variety of viral infection models [17,18].

While the effect of commensal microbe-derived metabolites on cellular activities and pathogen infections is clear, a reciprocal impact of the commensal microbiota and viral pathogens on metabolites has yet to be investigated. We previously showed that some commensal bacteria, including *Lactobacillus murinus*, significantly enhanced virally-induced leukemia development without affecting virus replication [19]. We used a similar model coupled with an unbiased metabolomics approach to investigate the influence of cross-talk between commensal bacterial and viral infection on metabolites. Here, we show viral infection and commensal bacterial colonization dependently alter the metabolomic profile in a sterile organ, the spleen.

## 2. Materials and Methods

### 2.1. Mice

Breeding and maintenance of mice used in this study were conducted at the animal facility of The University of Chicago. BALB/cJ mice were bought from The Jackson Laboratory (TJL). Females and males were used at a ratio of ~50:50. The Animal Care and Use Committee at The University of Chicago reviewed and approved the studies described here.

### 2.2. Monitoring Sterility in GF Isolators

BALB/cJ mice were re-derived as germ-free (GF) at Taconic farms and subsequently housed in sterile isolators at the gnotobiotic facility at the University of Chicago. GF isolator sterility was determined as previously described [20]. Fecal pellets from isolators were collected weekly and quickly frozen. A bead-beating/phenol-chloroform protocol was used to extract DNA. DNA was amplified using primers that widely hybridize to bacterial 16S rRNA gene sequences. In addition, microbiological cultures were established with GF fecal pellets, specific-pathogen-free (SPF) fecal pellets (positive control), sterile saline (sham), and sterile culture medium (negative control). Samples were inoculated into BHI, Nutrient, and Sabbaroud Broth media and incubated aerobically and anaerobically at 37 °C and 42 °C. Cultures were maintained for five days until deemed negative.

### 2.3. Colonization of GF Mice

*Lactobacillus murinus* (*L. murinus* ASF361) was isolated from the Altered Schaedler Flora (ASF) consortium [21]. Accordingly, fecal matter from ASF-colonized gnotobiotic mice was suspended in sterile PBS and plated onto a selective media for Lactobacilli, namely de Man, Rogosa, and Sharpe agar (MRS) (ThermoFisher Scientific, Waltham, MA, USA). The identity of *L. murinus* was confirmed by sequencing of the 16S RNA ribosomal amplicon generated via PCR from a bacteria colony. GF BALB/cJ mice were monocolonized with *L. murinus* by oral gavage of 200 μL of overnight liquid culture grown from a single colony. Monocolonization was validated by sequencing PCR products generated from fecal DNA and 16S rRNA primers and was verified at closing of the experiment.

### 2.4. Virus Isolation and Infection

Rauscher-like murine leukemia virus (RL-MuLV) is composed of N, B tropic ecotropic and mink cell focus forming (MCF) virus [22]. RL-MuLV was isolated from tissue culture supernatant of chronically infected SC-1 cells (ATCC CRL-1404). Titers of ecotropic virus within the RL-MuLV mixture were determined via an XC plaque assay [23]. 1 × 10^3^ pfus were intraperitoneally (i.p.) injected into 6–8-week-old BALB/cJ female mice (G0 mice). G0 mice were bred to produce the progeny G1 mice. Spleens from G1 mice were used as a source of virus. Homogenized spleens from 2–3-month-old G1 mice were centrifuged at 4 °C for 15 min at 2000 rpm. Supernatant was collected and layered onto a 30% PBS/sucrose cushion and spun at 31,000 rpm for 1 h at 4 °C in a TW55.1 bucket rotor. The pellet fraction was re-suspended in PBS. Insoluble material was removed by spinning the resuspended fraction at 4 °C at 10,000 rpm. Supernatant was collected and aliquoted, titered via plaque assay, and stored at −80 °C.

RL-MuLV isolated from spleens was diluted in sterile PBS followed by filtering through a sterile 0.22 μm membrane in a laminar flow hood. GF and *L. murinus*-monocolonized BALB/cJ females were injected i.p. with 1 × 10^3^ pfus (G0 mice). G0 females were bred to produce G1 mice, which were used for the metabolomic analysis. Uninfected and infected GF and *L. murinus*-monocolonized mice were sacrificed at 2 months. Plasma and spleens were isolated and stored at −80 °C. Mice were confirmed to be infected via PCR using primers specific for the LTR of ecotropic virus. Forward primer: 5′ATGAACGACCCCACCAAGT3′ and reverse primer: 5′GAGACCCTCCCAAGGAACAG3′. Spleen weights ranged from 0.07 to 0.09 g among uninfected GF mice; 0.07–0.09 g among uninfected *L. murinus*-monocolonized; and 0.15–0.19 g among infected *L. murinus*-monocolonized. Infected mice were non-leukemic as evaluated by histology of H&E-stained spleen sections.

### 2.5. FITC Permeability Assay

Permeability of the mouse gut was assessed using a FITC permeability assay. Food and water were removed from mouse cages for four hours. After four hours, mice were gavaged with 60 mg of FITC-dextran (MW 4000, Sigma-Aldrich, St. Louis, MO, USA) per 100 g of mouse [24]. Three hours post gavage, mice were bled into 100 μL heparin, and plasma was separated by spinning the blood at 2000 rpm for ten minutes at 4 °C. Then, 50 μL of each sample was loaded into a 96-well plate in duplicate to measure FITC concentration at emission and excitation wavelengths of 520 nm and 490 nm, respectively, using a TECAN fluorescence spectrophotometer and Magellan software. FITC-dextran standards were diluted in plasma from unmanipulated mice. Fluorescence from the negative control samples (plasma from unmanipulated mice) was subtracted from fluorescence of the standards and experimental samples.

### 2.6. Metabolomics

Spleens and sera from six GF uninfected, five GF infected, seven colonized uninfected, and twenty-two colonized infected mice were harvested and maintained at −80 °C prior to metabolite extraction. About 10 mg per sample was used for the following analysis. Extraction solvent (1:2:2 water:acetonitrile:methanol containing stable isotope-labeled internal standards) was added to each spleen sample at a ratio of 400 µL per 10 mg of tissue. Four to six 2.3 mm stainless steel beads were added to each sample tube and method blanks. Samples were homogenized by bead beating at 20 Hz for 15 min. Homogenized samples were placed at −20 °C for 1 h to maximize protein precipitation, followed by brief vortexing and centrifugation for 5 min at 4 °C and 14,000× *g*. Next, 120 µL of supernatant was taken and passed through a 0.2 µm polyvinylidene fluoride centrifugal filter at 4 °C for 3 min at 6000× *g*. Collected extract was stored at 4 °C during analysis.

Metabolites were extracted from serum samples as follows. Frozen specimens were thawed on ice and inverted five times to mix. From each specimen, 50 µL of serum was aliquoted into new 1.5 mL Eppendorf tubes. Then, 150 µL of 1:1 acetonitrile:methanol, containing stable isotope-labeled internal standards, was added to each 50 µL aliquot, followed by vortexing for 20 s. All tubes were stored at −20 °C for 1 h and then were promptly vortexed for 20 s and centrifuged at −10 °C for 10 min at 21,130× *g*. Supernatant was transferred to a new 1.5 mL Eppendorf tube and dried at room temperature in a LabConco Speedvac. Dried extracts were resuspended in 50 µL 4:1 acetonitrile:water and stored at 4 °C during analysis. Technical replicates of serum samples were made in the same fashion as described for spleen samples.

Data processing, chromatography, and tandem mass spectral data collection methods have been previously described elsewhere [25]. Briefly, hydrophilic interaction liquid chromatography (HILIC) method was used for analysis of the polar metabolites. Prepared samples were injected onto a Waters Acquity UPLC BEH Amide column (150 mm length × 2.1 mm id; 1.7 μm particle size) with an additional Waters Acquity VanGuard BEH Amide pre-column (5 mm × 2.1 mm id; 1.7 μm particle size) maintained at 45 °C coupled to an Thermo Vanquish UPLC. The mobile phases were prepared with 10 mM ammonium formate and 0.125% formic acid in either 100% LC-MS-grade water for mobile phase (A) or 95:5 *v*/*v* acetonitrile:water for mobile phase (B). Gradient elution was performed from 100% (B) at 0–2 min to 70% (B) at 7.7 min, 40% (B) at 9.5 min, 30% (B) at 10.25 min, and 100% (B) at 12.75 min, isocratic until 16.75 min with a column flow of 0.4 mL/min. Spectra were collected using a Thermo Q Exactive HF Hybrid Quadrupole-Orbitrap mass spectrometer in both positive and negative mode ionization (separate injections). Full MS-ddMS2 data were collected, and an inclusion list was used to prioritize MS2 selection of metabolites from the in-house “local” library; when additional scan bandwidth was available; MS2 data were collected in a data-dependent manner. The mass range was 60–900 mz, resolution was 60 k (MS1) and 15 k (MS2), centroid data were collected, loop count was 4, and isolation window was 1.2 Da.

Noise was defined as peak height below 100,000. Aligned peaks were retained if present in at least 10% of all samples and at least 60% of samples within at least one treatment group. Blank samples, generated by extracting water in place of biological material, were used to remove features not originating from serum or spleen. To be retained, all signals were required to have at least 10-fold increase in signal in one or more samples compared to the blank average. All annotations were assigned based on, at minimum, accurate mass and MS2 spectral matching against the largest freely available spectra repository, MassBank of North America [26].

Technical replicates from each sample matrix (spleen and serum, independently) were generated by pooling all samples of each matrix. Metabolites that were not analytically reproducible based on >30% reproducibility standard deviation in the technical replicates were removed prior to statistical analysis. Metabolites with RSDs less than 30% were normalized with Log_10_, quantile normalization, followed by determining the z-score using ANOVA.

## 3. Results

Bacterial colonization and viral infection drastically alter the peripheral metabolite landscape. To identify metabolites whose presence and abundance were dictated by both colonization and viral infection, an unbiased metabolomics approach was undertaken. Accordingly, we compared metabolites within the sera and spleens of murine leukemia virus (MuLV)-infected and uninfected BALB/cJ mice that were either germ-free (GF) or monocolonized with mouse commensal *Lactobacillus murinus* (*L. murinus*). We used Rauscher-like MuLV variant, which is spread via blood and replicates within the proliferating erythroid progenitor cells, subsequently causing erythroid leukemia [22]. GF and *L. murinus*-colonized ex-GF mice were injected with 0.22 μM sterilized virus and bred to produce offspring (G1 mice) infected via a natural route. Sera and spleens from four groups of 2-month-old G1 mice, namely uninfected GF, ex-GF *L. murinus*-colonized, MuLV-infected GF, and MuLV-infected ex-GF *L. murinus*-colonized, were subjected to coupled liquid chromatography/tandem mass spectrometry (LC-MS/MS). Mass spectrometry spectra were collected under positive and negative ionization modes. Metabolites whose frequencies had a relative standard deviation (RSD) greater than 30% between technical replicates were discarded, and the remaining metabolites were subjected to further analyses (Figure 1A–C).

Serum samples collected from uninfected and infected germ-free mice (Figure 1A) were subjected to LC-MS/MS separately from serum samples of uninfected and infected colonized mice (Figure 1B); therefore, it was impossible to match unknown metabolites between these two groups. Consequently, unknown metabolites from the sera only were discarded in later comparative analyses. Only identifiable metabolites found between all four groups of mice were kept for subsequent analyses. This includes 73 and 55 known metabolites identified under positive and negative ionization, respectively (Figure 1A,B and Appendix A). Z-scores for known metabolites can be found in Spreadsheet S1. MuLV infection of GF mice marginally altered the overall metabolomic profile (Figure 2A) and the level of metabolites within the sera (Figure 2B). In contrast, *L. murinus* colonization led to the greatest shift in metabolites (Figure 2A,B). MuLV infection of *L. murinus*-colonized mice induced further minimal changes in the overall metabolomic landscape and quantity of metabolites within the sera (Figure 2A,B).

From the spleens of all mouse groups, 60 known metabolites in total were identified under positive ionization and 52 under negative ionization (Figure 1C and Appendix A). Similar to the sera, the vast majority of changes were found among over 1500 unknown metabolites (Figure 1C). Contrary to the sera, splenic metabolites were altered most drastically in the presence of both *L. murinus* colonization and viral infection compared to either condition alone (Figure 2C,D). Therefore, viral infection in monocolonized mice led to negligible shifts in the metabolomic profile within the sera but greatly influenced metabolites within the spleen. These data indicate that bacterial colonization of the gut together with MuLV infection promotes metabolite accumulation or production in the spleen.

Through our analysis, we identified three groups of metabolites that, compared to GF mice, were altered upon MuLV infection or *L. murinus* colonization or both infection and colonization. Group 1 includes metabolites that were significantly altered in *L. murinus*-colonized mice; group 2 includes metabolites that were significantly altered in virally infected mice; and group 3 consists of metabolites that were significantly altered only in colonized and infected mice. As we were interested in metabolites whose presence and quantity were mediated by colonization and viral infection, we further investigated metabolites identified in group 3. Considering bacterial colonization alone conferred the largest shift in the metabolomic landscape within the sera (Figure 3A), it was not surprising that only 29 identifiable metabolites were found to be significantly altered in the sera of colonized and infected mice compared to the sera from the other three groups of mice (Figure 3A). As sera from GF uninfected and infected mice were subjected to LC-MS/MS separately from the sera of monocolonized uninfected and infected mice, unknown metabolites from the sera could not be compared between these groups. In contrast, 271 known and unknown metabolites were found to be impacted by both bacterial colonization and viral infection in the spleen (Figure 3C,D). However, the vast majority of them were unknown, and only 21 metabolites were identifiable (Figure 3D).

Viral infection has been shown to alter the intestinal barrier, enabling microbial translocation [28,29]. Based on the observation that MuLV infection of colonized mice greatly enhanced the abundance of metabolites within the spleen, we sought to determine whether MuLV increased the permeability of the intestines. To address this possibility, intestinal permeability was compared between uninfected and infected, conventionally housed, specific-pathogen-free (SPF) mice. Mice were orally gavaged with MW 4000 FITC-dextran, a large molecule that will only cross from the intestine into the periphery if the intestinal barrier has been compromised, and fluorescence within the plasma was assayed four hours later. Concentration of FITC detected within the plasma was similar between uninfected and infected SPF mice, suggesting MuLV does not alter intestinal permeability (Figure 4). Thus, enhanced permeability of the intestinal barrier is not the cause of increased metabolites within the sera and spleen in infected mice.

## 4. Discussion

The commensal microbiota has been demonstrated to modulate replication and pathogenesis of viruses from various families [30,31,32,33]. Furthermore, commensal-derived metabolites have exhibited both pro- and anti-viral effects. Conversely, any impact of viral infection on microbially-derived metabolites remained unknown. LC-MS/MS metabolomics has been successfully utilized to identify metabolites [34,35,36]. Therefore, we took a similar approach to discover bacterially-derived or bacterially-dependent host-derived metabolites that are influenced by viral infection.

Abundances of 271 metabolites were found to be affected by both bacterial colonization and viral infection in the spleen (Figure 3C). The vast majority of these metabolites (250) are unknown, and consequently, their relationship to known metabolites remains to be determined. Known metabolites within the sera and spleen, influenced by the combination of bacterial colonization and viral infection, were grouped by chemical type in an attempt to identify trends in the abundances of similar metabolites (Figure 3B,D). The only discernable trend determined by both the virus and bacterium was the decreased abundance of cholines in the sera and increased abundance of amino acids (both essential and non-essential) in the spleen (Figure 3D).

Mechanistically, how MuLV infection in *L. murinus*-colonized mice alters the metabolomic landscape in the spleen remains to be determined. MuLV infection results in increased extramedullary hematopoiesis within the spleen [22], expanding the population of target cells the virus can infect. MuLV readily replicates within the proliferating erythroid progenitor cells, augmenting the chance of pro-viral integration near a cellular proto-oncogene and leading to the generation of pre-cancerous cells. The presence of the microbiota during the generation of these cells may alter the transcriptional landscape, leading to the production of metabolites within the spleen.

Permeability of the intestinal barrier did not increase in SPF mice upon MuLV infection (Figure 4), indicating the observed increase of metabolites within the spleens of monocolonized infected mice may not be due to significant influx of metabolites originating from the gut. Thus, the origin of certain microbially dependent, virally induced metabolites may be within the spleen. However, due to the diffusible nature of small, gut-derived metabolites, we cannot rule out the gut as a source of these metabolites.

In summary, our study has demonstrated for the first time that the gut commensal bacterium and the virus together alter the metabolomic landscape of the spleen, a sterile organ not directly connected with the gut.

## Figures and Tables

**Figure 1 viruses-15-00386-f001:**
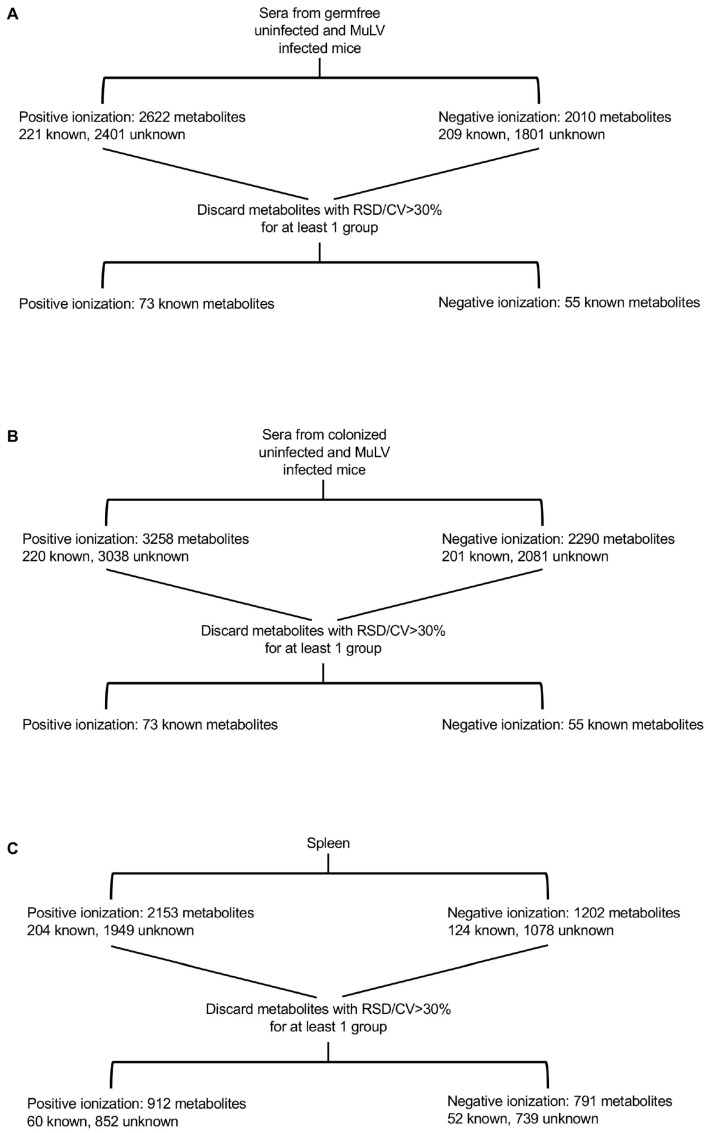
Flow chart illustrating the processing of sera and spleen samples for metabolomics. Mass spectrometry for serum samples was conducted in two separate procedures (**A**,**B**). Therefore, unknown metabolites could not be matched between the experiments and were removed. Spleen (**C**) samples were subjected to mass spectrometry under positive and negative ionization modes. Metabolites within a cohort of mice with a relative standard deviation (RSD) > 30% between technical replicates were discarded, leaving the indicated number of known and unknown metabolites.

**Figure 2 viruses-15-00386-f002:**
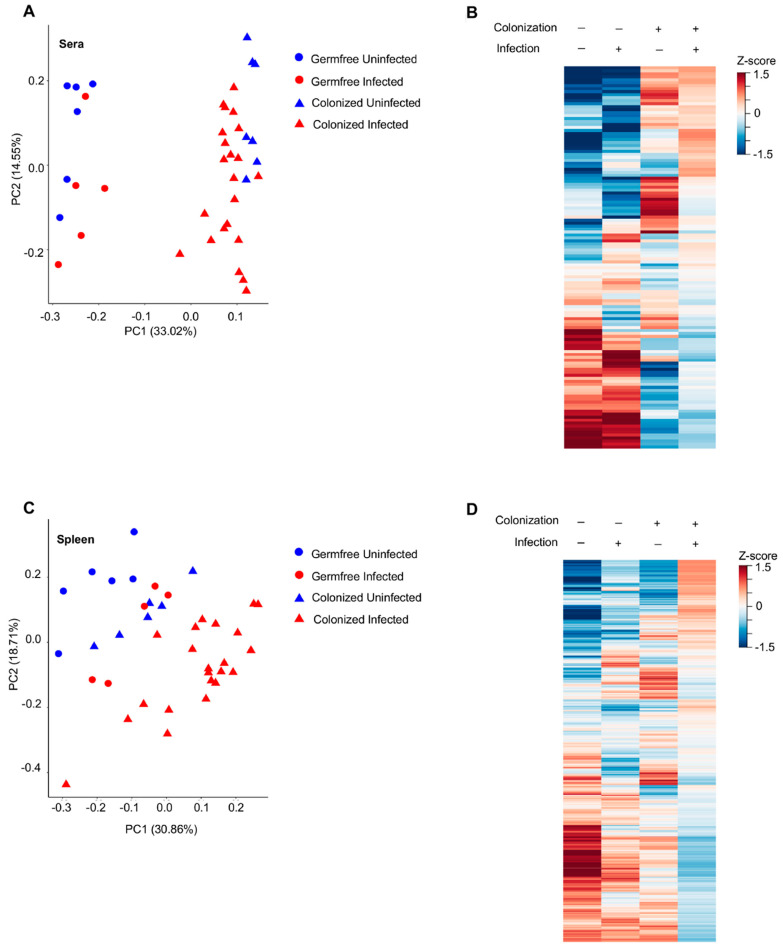
MuLV infection and bacteria colonization alter the metabolomic profile in the sera and spleen. (**A**, **C**) PCoA plot of all metabolites within the sera (**A**) or spleen (**C**) of the indicated mice. Each point is an individual mouse. (**B**,**D**) Heatmaps of metabolite abundances in the sera (**B**) or spleen (**D**) of the four groups of mice. Each row is a metabolite.

**Figure 3 viruses-15-00386-f003:**
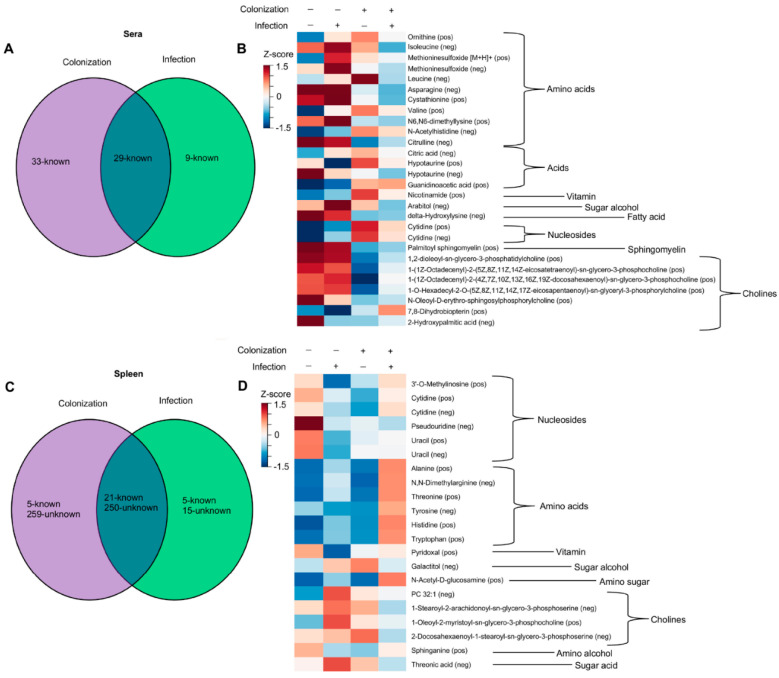
Abundances of certain metabolites is altered by colonization and MuLV infection. (**A**,**C**) Venn diagrams illustrating the numbers of metabolites significantly altered by colonization, infection, or both in the sera (**A**) or spleen (**C**). (**B**,**D**) Heatmaps of identified metabolites, clustered by chemical type, whose abundances are influence by colonization, MuLV infection, or both in the sera (**B**) or spleen (**D**). Normalized values for each group were averaged and constructed into a heat map where red color indicates high metabolite levels, and blue color indicates lower metabolite levels. *p*-values were adjusted using Bonferroni correction [27] followed by two-way ANOVA to identify metabolites influenced by two independent variables: the presence of *L. murinus* and the virus. Each row is a metabolite.

**Figure 4 viruses-15-00386-f004:**
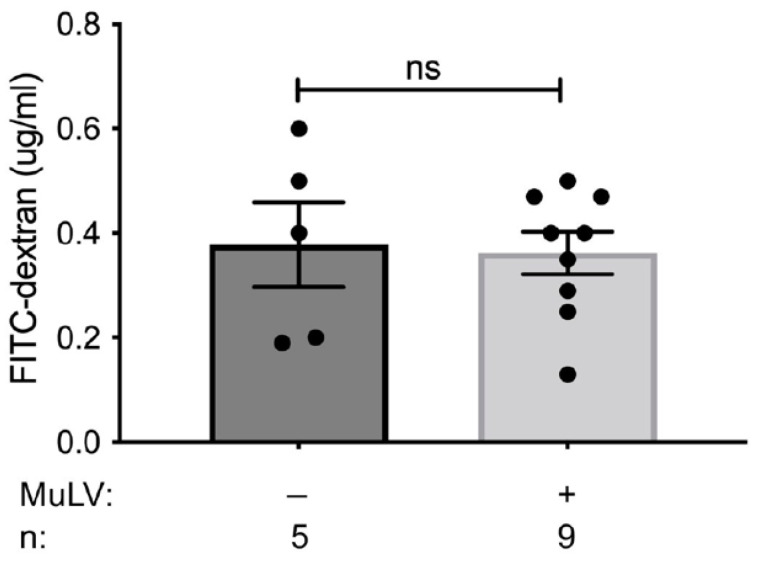
Intestinal permeability is unaffected by MuLV infection. FITC permeability assay was used to determine the permeability of the intestines in SPF BALB/cJ infected and uninfected mice. Each data point is an individual plasma sample. n, number of mice used. *p*-value calculated using unpaired *t*-test. ns, not significant.

## Data Availability

The data presented in this study are available on request from the corresponding author.

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
