# Peer review of "Retroviral Infection and Commensal Bacteria Dependently Alter the Metabolomic Profile in a Sterile Organ"

_viruses, 2023, doi:10.3390/v15020386_

Round 1

Reviewer 1 Report

The data presented indicate that there are metabolites detected from changes in microbiome resulting from either controlled lactobacillus infection, MuMLV infection, or both together. The results presented describe the metabolites that were detected in sera and spleen. There are some comments for the authors to address:

1. Please provide rationale for discarding data that were >30% from relative standard deviation. How was that level chosen and is that considered justifiable in the field?

2. Can the metabolites be categorized in some way so that conclusions can be drawn about the significance of the findings? As they stand, they are interesting but the importance is not evident from the way the information was presented

3. Maybe it was stated and I missed it--What is the source of the metabolic changes in the MuMLV infected mice? Were they directly related to bacterial  replication or resulting from cellular metabolism, cell proliferation, or cell death? This section of the results/discussion was unclear.

Author Response

The data presented indicate that there are metabolites detected from changes in microbiome resulting from either controlled lactobacillus infection, MuMLV infection, or both together. The results presented describe the metabolites that were detected in sera and spleen. There are some comments for the authors to address:

  1. Please provide rationale for discarding data that were >30% from relative standard deviation. How was that level chosen and is that considered justifiable in the field?

Technical replicates from each sample matrix (spleen and serum, independently) were generated by pooling all samples of each matrix. Metabolites which were not analytically reproducible, based on >30% reproducibility standard deviation in the technical replicates, were removed prior to statistical analysis. We have clarified in the text that >30% standard deviation is the cutoff for technical replicates only. Lines 172-176

Can the metabolites be categorized in some way so that conclusions can be drawn about the significance of the findings? As they stand, they are interesting but the importance is not evident from the way the information was presented.

We categorized the metabolites by chemical type (Figures 3B and 3D). This revealed an increase of amino acids and a decrease of cholines in the spleens of infected/colonized mice compared to the other three groups of mice. Metabolites whose abundances are dependent on viral infection and colonization were placed into metabolomic pathways. However, so far, we were unsuccessful in identifying pathways that could connect them. While we agree that the presented material is rather descriptive, we would like to emphasize that the results of these investigates introduce us to the new realm of metabolic interplay between viruses and bacteria. With the progress of metabolomics and development of new analytical tools, it should become clear that the pioneering studies (like this one) of viral-bacterial-host interactions were very important.

Maybe it was stated and I missed it--What is the source of the metabolic changes in the MuMLV infected mice? Were they directly related to bacterial  replication or resulting from cellular metabolism, cell proliferation, or cell death? This section of the results/discussion was unclear.

While it is clear viral infection and bacteria colonization interdependently alter metabolite abundances, the precise mechanism that induces these metabolomic changes remains to be determined. This is an important question; however, we feel it to be outside the scope of the current manuscript.

Reviewer 2 Report

In this manuscript, Spring et al. perform a metabolomic profile study following murine leukemia virus infection in germ-free mice and mice colonized with L. murinus bacteria. Authors report the metabolomic profile in both the blood sera and spleen of these animals. Authors conclude that the infection by MLV significantly changed the metabolomic profile of mice colonized with L. murinus. This manuscript is well written and should be of interest to the readers of Viruses. I listed some suggestions below to improve the manuscript. These are considered minor concerns as I don’t believe additional experimentation is necessary.

1. Authors choose the phrase “sterile organ” in both the title, the last sentence of the introduction and the discussion. I recommend actually using the word spleen for clarity.

2. The results reported in this study are mostly dependent on the metabolomic analysis, hence I highly recommend that authors at least briefly explain how they processed the data from mass spec instead of only citing a pre-print article. For example, Z-score and significance calculations would be useful for the readers.

3. Authors provided the known metabolites identified in this study in the supplementary materials, but I highly recommend that they also provide the matching Z-score data and/or the heat maps (match Fig 2B and D with supplementary) for each of these so that future researchers may utilize them.

4. In Figure 3A, authors only report known metabolites as those that are significantly altered in different conditions. But in Figure 3C they also report unknown metabolites. Are there no unknown metabolites significantly altered in the sera? Moreover in line 170 authors state that they discarded the unknown metabolites for comparative analysis however they report them in Figure 3C. This may create confusion for the readers

5. In Figure 3B, I see significant changes in color profile in most of the cholines upon colonization and upon infection of the colonized mice. I recommend authors point this out.

Author Response

Reviewer 2

In this manuscript, Spring et al. perform a metabolomic profile study following murine leukemia virus infection in germ-free mice and mice colonized with L. murinus bacteria. Authors report the metabolomic profile in both the blood sera and spleen of these animals. Authors conclude that the infection by MLV significantly changed the metabolomic profile of mice colonized with L. murinus. This manuscript is well written and should be of interest to the readers of Viruses. I listed some suggestions below to improve the manuscript. These are considered minor concerns as I don’t believe additional experimentation is necessary.

  1. Authors choose the phrase “sterile organ” in both the title, the last sentence of the introduction and the discussion. I recommend actually using the word spleen for clarity.

Thank you for the comment. We have made appropriate changes.

  1. The results reported in this study are mostly dependent on the metabolomic analysis, hence I highly recommend that authors at least briefly explain how they processed the data from mass spec instead of only citing a pre-print article. For example, Z-score and significance calculations would be useful for the readers.

Thank you for bringing this to our attention. We have added more detailed text to the materials and methods regarding LC-MS/MS conditions, how data was processed, and how metabolites were annotated. Lines 141-171.

  1. Authors provided the known metabolites identified in this study in the supplementary materials, but I highly recommend that they also provide the matching Z-score data and/or the heat maps (match Fig 2B and D with supplementary) for each of these so that future researchers may utilize them.

Z-score data for the known metabolites described in Figures 2B and 2D are now included as Spreadsheet S1.

  1. In Figure 3A, authors only report known metabolites as those that are significantly altered in different conditions. But in Figure 3C they also report unknown metabolites. Are there no unknown metabolites significantly altered in the sera? Moreover in line 170 authors state that they discarded the unknown metabolites for comparative analysis however they report them in Figure 3C. This may create confusion for the readers

There were unknown metabolites found in the sera. However, as sera isolated from the different groups of mice were subjected to LC-MS/MS separately, the unknown metabolites found could not be compared between the groups. This has now been reiterated and clarified. As to your second point, in line 170, we have change “unknown metabolites from the sera have been discarded” to “unknown metabolites from the sera onlywere discarded” (Figure 3A). Lines 247-249.

  1. In Figure 3B, I see significant changes in color profile in most of the cholines upon colonization and upon infection of the colonized mice. I recommend authors point this out.

Thank you for the comment. This has now been included in the discussion.

Reviewer 3 Report

Spring et al. shows in this manuscript that MuLV infection alters the metabolomic profile in the absence or presence of Lactobacillus murinus. The purpose of this study is very interesting, but the  several control experiments are missing. Therefore, the authors cannot conclude so. Specific comments are described: 

1. How many mice per each group were used?

2. "G0 females were bred to produce G1" (line 93). In this study, G0 or G1 mice were analyzed? 

3. The authors have already reported that L. murinus enhances pathogenesis of MLV (19). Thus, L. murinus may facilitate the MLV replication in vivo. Viral titers should be compared between in the absence and presence of L. murinus, when the samples were isolated.

4. There are many reports showing that metabolomic profile is changed by transformation. Although leukemic cells were not observed by histology, spleen weight was clearly higher in infected-mice. So, the possibility that the metabolomic profile is altered by activation of spleen cell growth, but not by MLV infection itself, cannot be excluded. Thus, the conclusion "retroviral infection alters the metabolomic profile" should be changed to "retrovirus-mediated cellular activation alters the metabolomic profile". In order to analyze the impact of MLV infection on metabolomic profile, it should be measured when spleen weights of infected and uninfected mice are similar. 

Author Response

Reviewer 3

Spring et al. shows in this manuscript that MuLV infection alters the metabolomic profile in the absence or presence of Lactobacillus murinus. The purpose of this study is very interesting, but the several control experiments are missing. Therefore, the authors cannot conclude so. Specific comments are described: 

  1. How many mice per each group were used?

Sera and spleens from six GF uninfected, five GF infected, seven colonized uninfected, and 22 colonized infected mice were used for the metabolomic analysis. This has been added to the materials and methods. Lines 119-120.

  1. "G0 females were bred to produce G1" (line 93). In this study, G0 or G1 mice were analyzed? 

G1 mice were analyzed. This has now been specified. Line 97.

The authors have already reported that L. murinus enhances pathogenesis of MLV (19). Thus, L. murinus may facilitate the MLV replication in vivo. Viral titers should be compared between in the absence and presence of L. murinus, when the samples were isolated.

The same publication demonstrates L. murinus does not enhance MuLV replication compared to GF mice as we compared the virus load in L. murinus colonized, GF and SPF mice. This clarification has been added to the current manuscript. Line 47.  Even though we did not perform an infectious center (IC) assay with the spleens we used for metabolomics studies (due to the limited material), we performed the IC assay with spleens from different MuLV-infected L. murinus-colonized ex-germ free and germ-free G1 mice at 2 months of age (the same age we used mice for the studies) and saw no difference in the frequency of infected cells (Figure 1 for Reviewers). 

  1. There are many reports showing that metabolomic profile is changed by transformation. Although leukemic cells were not observed by histology, spleen weight was clearly higher in infected-mice. So, the possibility that the metabolomic profile is altered by activation of spleen cell growth, but not by MLV infection itself, cannot be excluded. Thus, the conclusion "retroviral infection alters the metabolomic profile" should be changed to "retrovirus-mediated cellular activation alters the metabolomic profile". In order to analyze the impact of MLV infection on metabolomic profile, it should be measured when spleen weights of infected and uninfected mice are similar. 

Hundred percent of infected mice show enlarged spleens as a result of the extramedullary hematopoiesis (EMH) induced by the virus (1). However, both infected GF and colonized mice exhibited similar expansion of hematopoietic stem cells (2) suggesting that changes in metabolic profiles uniquely found in infected colonized mice cannot be explained by proliferation of hematopoietic stem cells.

References

  1. L. M. Hook et al., Characterization of a novel murine retrovirus mixture that facilitates hematopoiesis. Journal of virology 76, 12112-12122 (2002).
  2. J. Spring et al., Gut commensal bacteria enhance pathogenesis of a tumorigenic murine retrovirus. Cell Reports 40, (2022).

Reviewer 4 Report

This is a descriptive study without any mechanistic analyses.  There are numerous inconsistent descriptions and it is unclear if the detected metabolites are derived from microorganisms.

Major concerns

1) In lines 240-242 the authors describe that they took an approach to discover bacterially derived or bacterially-dependent host-derived matabolites that are influenced by viral infection, but they actually made no distinctions between bacterially derived and host-derived metabolites.  In this regard, amino acids are uniformly decreased in sera of bacterially colonized and virus-infected mice while they are highly increased in the spleen.  Dose this mean that serum amino acids are consumed by the proliferation of infected cells in the spleen?

2) The title says retroviral infection and commensal bacteria "dependently" alter the matabolic profile, but is there any evidence that indicates the interdependence in the results?  Further, the title also indicates the spleen is sterile in the subject animals.  However, what is shown in the results is the lack of FITC-dextran leakage in separate SPF animals, and no demonstration of the sterility of spleens in colonized and virus-infected animals is shown.

3) In line 176, the changes in matabolic landscape in the colonized and infected animals are described to be minimal, but actual changes are distinctive both in Figures 2 and 3, especially regarding amino acids.

4) Hyphenation and use of italics are inconsistent.  In the Introduction, "bacteria derived" and "microbe-derived" are both used.  Similarly, "microbially derived" and "bacterially-derived" are both found in the Discussion section.  Lactobacillus murinus or L. murinus are not italicized in sections 2.3, 2.4 but are properly italicized in sections 3 and 4.

Minor problems

1) Descriptions on virus infection procedures in lines 82-85 and those in lines 92-94 are repetitive.

2) Sera samples must be serum samples or sample sera.

Author Response

Reviewer 4

This is a descriptive study without any mechanistic analyses.  There are numerous inconsistent descriptions and it is unclear if the detected metabolites are derived from microorganisms.

Major concerns

  • In lines 240-242 the authors describe that they took an approach to discover bacterially derived or bacterially-dependent host-derived matabolites that are influenced by viral infection, but they actually made no distinctions between bacterially derived and host-derived metabolites.  In this regard, amino acids are uniformly decreased in sera of bacterially colonized and virus-infected mice while they are highly increased in the spleen.  Dose this mean that serum amino acids are consumed by the proliferation of infected cells in the spleen?

While it would be very interesting to know if the decrease of amino acids in the serum is related to the increase of amino acids in the spleen and whether this is related to cellular metabolism, we believe this question to be outside of the scope of this manuscript.

  • The title says retroviral infection and commensal bacteria "dependently" alter the matabolic profile, but is there any evidence that indicates the interdependence in the results?  Further, the title also indicates the spleen is sterile in the subject animals.  However, what is shown in the results is the lack of FITC-dextran leakage in separate SPF animals, and no demonstration of the sterility of spleens in colonized and virus-infected animals is shown.

As the abundances of many metabolites in the sera and spleen were altered in only colonized and infected mice compared to colonized or GF infected mice (Figures 3B and 3D), we believe these results indicate an interdependence of L. murinus and MuLV on these metabolites.

Spleen is considered to be a sterile organ even in SPF mice.  We have performed PCR on 16S bacterial ribosomal RNA genes to demonstrate that spleens from SPF mice has no bacterial DNA (Figure 2 for Reviewers).

  • In line 176, the changes in matabolic landscape in the colonized and infected animals are described to be minimal, but actual changes are distinctive both in Figures 2 and 3, especially regarding amino acids.

While MuLV infection of colonized mice does induce changes in the abundances of some metabolites within the sera, the overall changes to the metabolomic landscape appear to be minimal. This has been edited to reflect this clarification. Line 219.

  • Hyphenation and use of italics are inconsistent.  In the Introduction, "bacteria derived" and "microbe-derived" are both used.  Similarly, "microbially derived" and "bacterially-derived" are both found in the Discussion section.  Lactobacillus murinus or L. murinus are not italicized in sections 2.3, 2.4 but are properly italicized in sections 3 and 4.

Thank you for the comment. Edits have been made to be more consistent.

Minor problems

  • Descriptions on virus infection procedures in lines 82-85 and those in lines 92-94 are repetitive.

Lines 82-85 describe the process of isolating virus for infection. Lines 92-94 describe the infection protocol for mice used in the analysis.

  • Sera samples must be serum samples or sample sera.

This has been corrected.

Round 2

Reviewer 3 Report

The authors said in the Introduction section that metabolites promote the immune response and augment IFN expression in a variety of viral infection models. The relationship between metabolites and pathogens is mentioned. It means that the authors wanted to show the enhancement of the MuLV pathogenesis by changes in a metabolite profile. However, if "the both infected GF and colonized mice exhibited similar expansion of hematopoietic stem cells (no change in the MuLV pathogenesis)", the purpose of this study is unclear. If the authors wanted to show the interaction between the virus and bacteria in  metabolism, they should mainly describe about their relationship in the Introduction section, and measure the number of L. murinus in the MuLV-infected and uninfected mice.

Author Response

R2

The authors said in the Introduction section that metabolites promote the immune response and augment IFN expression in a variety of viral infection models. The relationship between metabolites and pathogens is mentioned. It means that the authors wanted to show the enhancement of the MuLV pathogenesis by changes in a metabolite profile. However, if "the both infected GF and colonized mice exhibited similar expansion of hematopoietic stem cells (no change in the MuLV pathogenesis)", the purpose of this study is unclear. If the authors wanted to show the interaction between the virus and bacteria in  metabolism, they should mainly describe about their relationship in the Introduction section, and measure the number of L. murinus in the MuLV-infected and uninfected mice.

We are sorry for not being clear. Extramedullary hematopoiesis (EMH) is a result of MuLV-induced proliferation of hematopoietic stem cells (HSCs) in the spleen and occurs in 100% of MuLV-infected SPF and GF mice (1). This phase in disease development is preceding leukemia. Actively proliferating cells is a reservoir of infection-competent cells since the retrovirus, such as MuLV requires dividing cells for productive infection. In some infected cells proviral integrations occur next to the cellular protooncogenes thus, upregulating them. These cells give rise to clonal leukemia. Some commensal bacteria, including Lactobacillus murinus significantly enhanced virally-induced leukemia development without affecting EMH required for virus replication (2). WE clarified this in lines 44-47.

Additionally, viral infection does not change L. murinus load in the gut (Figure 1 for R2).

References

  1. L. M. Hook et al., Characterization of a novel murine retrovirus mixture that facilitates hematopoiesis. Journal of virology 76, 12112-12122 (2002).
  2. J. Spring et al., Gut commensal bacteria enhance pathogenesis of a tumorigenic murine retrovirus. Cell Reports 40, (2022).
